# Prostate-specific antigen (PSA) testing of men in UK general practice: a 10-year longitudinal cohort study

Grace J Young,[1,2] Sean Harrison,[1] Emma L Turner,[1] Eleanor I Walsh,[1] Steven E Oliver,[3] Yoav Ben-Shlomo,[1] Simon Evans,[4] J Athene Lane,[1,2] David E Neal,[5] Freddie C Hamdy,[5] Jenny L Donovan,[1] Richard M Martin,[1] Chris Metcalfe[1,2]

GJY and SH contributed equally.

[1]Population Health Sciences, Bristol Medical School, University of Bristol, Bristol, UK
[2]Bristol Randomised Trials Collaboration, University of Bristol, Bristol, UK
[3]Health Sciences, Hull York Medical School, University of York, York, UK
[4]Urology Department, Royal United Hospital, Bath, UK
[5]Nuffield Department of Surgical Sciences, University of Oxford, Oxford, UK

**Correspondence to**
Professor Chris Metcalfe;
chris.metcalfe@bristol.ac.uk

## ABSTRACT

**Objectives** Cross-sectional studies suggest that around 6% of men undergo prostate-specific antigen (PSA) testing each year in UK general practice (GP). This longitudinal study aims to determine the cumulative testing pattern of men over a 10-year period and whether this testing can be considered equivalent to screening for prostate cancer (PCa).

**Setting, participants and outcome measures** Patient-level data on PSA tests, biopsies and PCa diagnoses were obtained from the UK Clinical Practice Research Datalink (CPRD) for the years 2002 to 2011. The cumulative risks of PSA testing and of being diagnosed with PCa were estimated for the 10-year study period. Associations of a man's age, region and index of multiple deprivation with the cumulative risk of PSA testing and PCa diagnosis were investigated. Rates of biopsy and diagnosis, following a high test result, were compared with those from the programme of PSA testing in the Prostate Testing for Cancer and Treatment (ProtecT) study.

**Results** The 10-year risk of exposure to at least one PSA test in men aged 45 to 69 years in UK GP was 39.2% (95% CI 39.0 to 39.4%). The age-specific risks ranged from 25.2% for men aged 45–49 years to 53.0% for men aged 65–69 years (p for trend <0.001). For those with a PSA level ≥3, a test in UK GP was less likely to result in a biopsy (6%) and/or diagnosis of PCa (15%) compared with ProtecT study participants (85% and 34%, respectively).

**Conclusion** A high proportion of men aged 45–69 years undergo PSA tests in UK GP: 39% over a 10-year period. A high proportion of these tests appear to be for the investigation of lower urinary tract symptoms and not screening for PCa.

**Trial registration number** ISRCTN20141297,NCT02044172.

## Strengths and limitations of this study

► This is the first study in the UK to look at patterns of prostate-specific antigen (PSA) testing over a 10-year period in a cohort of men.
► Data on over 430 000 men could be analysed from the Clinical Practice Research Database and compared with data on 58 500 men from the programme of PSA testing and diagnostic biopsy in the Prostate Testing for Cancer and Treatment (ProtecT) study.
► The completeness of some routine data items is uncertain; with the recorded diagnoses outnumbering the recorded biopsies indicating that the latter are under-recorded.
► It was not possible to distinguish tests undertaken in men with and without symptoms; therefore, the proportion of tests prompted by the presentation of lower urinary tract symptoms was inferred.

test, with prostate biopsy in men with a raised PSA level allowing histopathological confirmation of the diagnosis of PCa. Despite almost 30 years of PSA testing, the balance of benefits and harms of the test has not been established and, perhaps as a consequence, there are varying rates of testing around the UK and the world.[2] There is evidence that a PSA-based screening programme will reduce mortality due to PCA[3] but with a risk of over diagnosis, such that a man diagnosed with cancer localised to the prostate would not have developed clinical symptoms of the disease in his lifetime if left untreated.[4][5] Radical treatment of such men exposes them to the risk of treatment-related adverse events without the potential to benefit.[6]

Current guidance for Primary Care Physicians in the UK, USA and Australia recommends discussing and coming to a shared decision about PSA testing,[7] with men who either raise the issue or warrant consideration of testing, due to a family history of the

## INTRODUCTION

The UK currently runs three screening programmes for breast, bowel and cervical cancers. Prostate cancer (PCa) is now the most commonly diagnosed cancer in men in the UK despite there being no formal screening programme.[1] Prostate-specific antigen (PSA) level can be used as a screening

**BMJ**

disease, for example. With such passive advice, variable testing rates across GPs are unsurprising. Three cross-sectional studies have been conducted giving an indication of the PSA testing rates in the UK between 2001 and 2011. Melia *et al*, studying 469 159 men aged 45–84 years, reported an annual rate of 6% over 1999–2002 for England and Wales, with an annual rate of 2% in the absence of symptoms.[8] Williams *et al*, studying 126 716 men aged 45–89 years and without a prior diagnosis of PCa, found that 6.2% of these men received a PSA test during 2007.[9] This study concluded that testing was more prevalent in older men, more southern areas of the UK (especially Wales) and areas of lower deprivation. Moss *et al* obtained data from the Clinical Practice Research Datalink (CPRD) on 650 264 men aged 45–84 years and found a testing rate of 8.74 and 9.45 per 100 person-years in 2010 and 2011, respectively.[10] Again, rates increased with age and areas of lower deprivation. Of 49 306 men tested in 2010 and with at least 9 months of follow-up, 0.2% of men with a PSA level <3 ng/mL were diagnosed with PCa within 9 months, rising to 14.5% of men with PSA level >5 ng/mL. A London-based study of 150 481 men aged 40 years or older found that 8.2% of men were PSA-tested at their general practice (GP) in 12 months from August 2013 to July 2014.[11]

When PSA tests are undertaken for screening, men with a raised level will be referred for biopsy, with examination of prostate tissue necessary for the diagnosis of PCa. Furthermore, as screen-detected PCa is relatively slow to progress, screening is targeted at men in their 50s and 60s, the balance of risks of short-term treatment harms and longer-term survival benefit being less favourable for older men as death due to other causes is more likely and radical treatments less suitable. Tests which are unlikely to be followed up by biopsy, and which are undergone by older men, are likely to be guiding the treatment of benign hyperplasia of the prostate.[12] Guidance for the assessment of lower urinary tract symptoms (LUTS), affecting approximately 30% of men over 50s,[13] includes consideration of a PSA when LUTS are suggestive of bladder outlet obstruction secondary to benign prostatic enlargement; where PSA >1.4 ng/mL can direct drug treatment decisions.[14]

While estimates of the number of men undergoing a PSA test in a 12-month period give an indication of how widespread use of the test has become in UK GP, a longitudinal perspective is needed to examine how the PSA test is being used to manage the risk of PCa in individual men. Long-term retrospective cohort studies of PSA testing rates have been conducted elsewhere in Europe[15 16]; however, the cumulative risks of PSA testing in the UK are yet to be quantified.

The primary objective of this study was to estimate the cumulative risk of PSA testing of UK men in primary care, without a diagnosis of PCa, over the 10-year period 1 January 2002 to 31 December 2011. The association of testing rates with age, region and index of multiple deprivation (IMD) was investigated. The proportion of tests resulting in a biopsy and/or diagnosis of PCa was compared with the programme of PSA testing, akin to screening, in the Prostate Testing for Cancer and Treatment (ProtecT) study[17] to gauge whether PSA tests undertaken in UK GP can be considered as an effective attempt at screening.

## SUBJECTS AND METHODS
### Design
We undertook a retrospective cohort study of 450 000 men using data from the CPRD, a large primary care database.[18] The CPRD contains electronic medical records for approximately 4.4 million active patients in 674 practices, representing 6.9% of the UK population. Patients in the database were shown to be representative of the UK population in terms of age, sex, ethnicity and body mass index. However, the data do not include prisoners, private patients, some residential homes and the homeless.[19] Practices participating in the CPRD have been found to have a greater number of patients compared with the national average.[20]

Data were requested for GP surgeries in all areas of UK but excluding London as it is thought that PSA testing rates would be markedly different in the capital.[11] We included practices which contributed acceptable 'research standard' data for the observation period, 1 January 2000 to 31 December 2011.[19] Data requested from the CPRD included: age, IMD from 2004, region, GP practice size, mortality date and cause, occurrence of PSA tests and prostate biopsies. PSA test dates before 2002 were also collected to estimate how many of the men had received a test prior to registration. IMD is an area-based deprivation measure which ranges from 0 to 100 with higher scores indicating higher levels of deprivation. CPRD bases these on the patients' postcode (English residents only) and then creates twentiles to ensure concealment of individuals' place of residence.

### Study population
Entry to the cohort commenced on 1 January 2002. Person-years for the time before the first PSA test were calculated having censored men from the analysis at the earliest of: (1) the end of the study period (31 December 2011), (2) after receiving a PCa diagnosis or (3) death or transfer out of the practice. Men aged 45 to 69 years at study entry were included (those born between 1933 and 1957).

Practices thought to be involved with research involving practice-wide PSA testing within the eligible age group were excluded. For example, the ProtecT study[17] was recruiting at UK GPs during 2001 to 2009. This exclusion was done by calculating the PSA testing rate for the men in each practice for all 60 2-month periods within the observation period and excluding a practice if in any 2-month period all the following conditions were satisfied: (1) the testing rate was >3.5 SDs higher than the overall practice average, (2) more than 10 men were tested and (3) more

than 5% of men not previously tested were tested in this period.

## Statistical analysis

The follow-up period for each man was calculated as the difference in years between registration start date (on or after 1 January 2002) and censoring (defined above). The Kaplan-Meier failure function estimated the cumulative proportion of men exposed to at least one PSA test and diagnosed with PCa over the course of the 10-year period for all men. The log-rank test was then used to investigate relationships between characteristics of the men and risk of undergoing a PSA test. A Cox proportional hazards model and Wald test were also used to check that associations remained, with or without accounting for clustering by practice. For men with full 10-year follow-up and no diagnosis of PCa (before or during follow-up), logistic and ordinal logistic regression were used to explore relationships between age group and the number of tests each man received.

The percentage of men retested within 365 days of their first test was explored by age category and PSA level for the CPRD data. For this analysis, all men in CPRD who had 365 days of follow-up post-PSA test were included; as well as those diagnosed within 365 days. Associations between age and retesting were investigated using logistic regression. Serum PSA levels for both the first and second test were used to determine the percentage of men diagnosed out of those who were retested, given their first and second PSA levels.

Data on men who attended PSA screening as part of the ProtecT study[17] were used to explore how routine data on PSA tests compare with the tests carried out as part of a screening intervention. The ProtecT data were divided between men with LUTS and no LUTS and compared with the CPRD dataset, by age group and further broken down by PSA level. LUTS were defined using five questions from the International Continence Society Male Short-Form questionnaire.[21] Men were classified as having LUTS if any of the following were true: (1) urinating every 2 hours or more during the day, (2) urinating at least twice during the night or (3) suffering 'sometimes', 'most of the time' or 'all of the time' from delayed urination, rushing to urinate or leaking before reaching the toilet. Men diagnosed with PCa before having their first PSA test in ProtecT were removed. PSA level was broken down into the following categories to ensure adequate numbers in each group: PSA <3, 3≤PSA<4, 4≤PSA<6, 6≤PSA<10 and PSA ≥10.

As an additional exploratory analysis, the percentage of men undergoing prostate biopsy and the percentage diagnosed with PCa within 365 days of their PSA test are also presented for men in the CPRD dataset and the two groups of ProtecT participants (LUTS and no LUTS). Comparisons between cohorts, and between risk groups within a cohort, were made using logistic and ordinal logistic regression. Within CPRD, biopsies and diagnoses were detected using medcodes provided by CPRD which

correspond to Read-codes which are used in GP in the UK. Lists used are in online supplementary table 1.

The CPRD group holds ethical approval from a National Research Ethics Service Committee for all purely observational research using anonymised CPRD data. The ProtecT trial holds ethics approval from the Trent Multicentre Research Ethics Committee (21/06/2001, ref: 01/4/025).

## RESULTS

### Final cohort

In total, 450 000 men from 578 primary care practices across all regions of the UK (excluding London) were included in the CPRD data extract. Of these 450 000 men: 14 were removed due to missing or conflicting data; 303 were listed as having died before 2002; 2184 had a diagnosis of PCa date before 2002 and 369 patients had no follow-up. From the remaining 447 130 patients, 12 894 (3%) men were removed as they were attendees of 19 practices suspected of participating in research involving practice-wide PSA testing. After the removal of these practices, the final sample was 434 236 men from 558 practices. Of these, 161 478 (37%) had the full 10-year follow-up.

### Risk of PSA testing and PCa diagnosis

The men were followed up for a cohort total of 2 963 645 person-years (median 8.25 years, IQR 3.83–10.00). Between 2002 and 2011 inclusively, 120 697 (28%) men received at least one PSA test, and 7538 (2%) men received a PCa diagnosis. The cumulative 1-year, 5-year and 10-year risks of receiving a PSA test were 5.1% (95% CI 5.0% to 5.2%), 21.4% (95% CI 21.3% to 21.5%) and 39.2% (95% CI 39.0% to 39.4%), respectively. The Kaplan-Meier curve illustrates the cumulative risk of PSA testing over the 10-year period, along with the age-specific cumulative risks (figure 1A). A similar trend was seen in PCa diagnoses (figure 1B). The cumulative 1-year, 5-year and 10-year risks of receiving a PCa diagnosis were 0.2% (95% CI 0.2% to 0.2%), 1.0% (95% CI 1.0% to 1.1%) and 2.7% (95% CI 2.7% to 2.8%), respectively.

Table 1 shows the risks by age group, region, IMD quartiles and testing history. The risk of receiving a PSA test for men in the lowest age category (45–49 years) was substantially lower than the highest age category (65–69 years), with 10-year risks of exposure to PSA testing of 25.2% and 53.0%, respectively (p<0.001). Likewise, the risk of diagnosis was also lower, with 10-year risks of 0.5% and 6.3% for age groups 45–49 years and 65–69 years, respectively.

PSA testing and diagnosis risks varied by region (p<0.001). The risks of testing and diagnosis were higher in more southern areas, especially the South East Coast (47.5% and 3.1%, respectively) and Wales (45.0% and 2.8%, respectively). The lowest risks were found in Scotland (23.8% and 2.4%, respectively) and the North East (30.5% and 2.3%, respectively). Those living in areas of greater deprivation had a lower risk of testing (46.3% vs 31.9%) and diagnosis (3.2% vs 1.9%); p for trend <0.001.

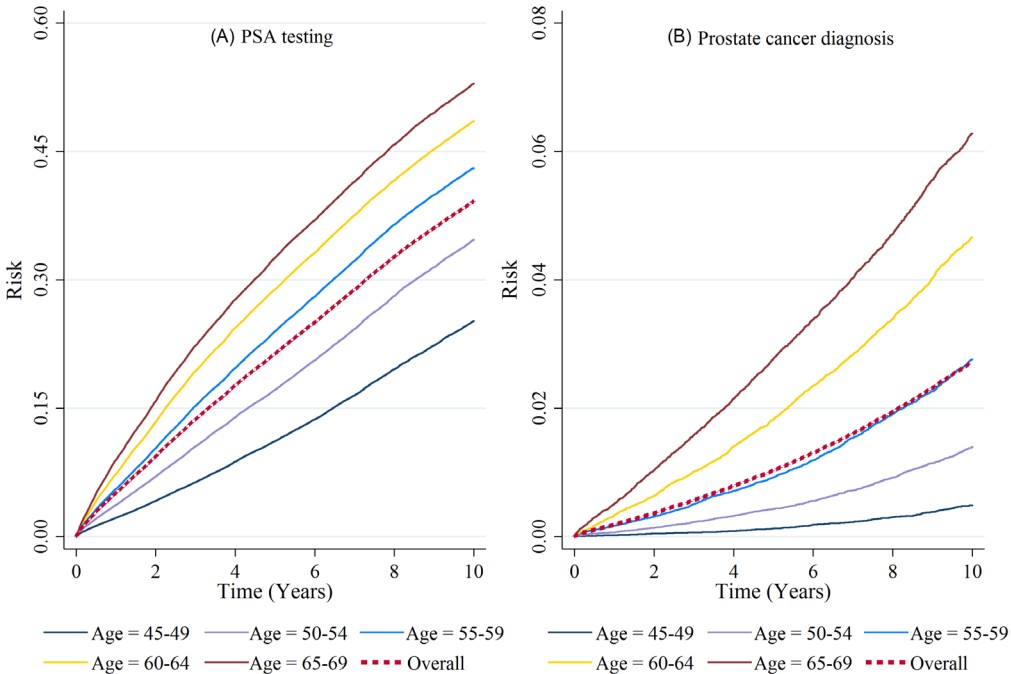

**Figure 1** (A) Kaplan-Meier failure estimate: cumulative risk over 10 years of receiving a PSA test, by age group, during the period 2002 to 2012. (B) Kaplan-Meier failure estimate: cumulative risk over 10 years of receiving a prostate cancer diagnosis, by age group, during the period 2002 to 2012. PSA, prostate-specific antigen.

Those who had received a PSA test prior to registration were substantially more likely to receive a PSA test and diagnosis than those who had not (73.2 vs 37.0 and 5.9 vs 2.5, respectively); p<0.001.

### Number of PSA tests
There were 157 586 men with complete 10-year follow-up and no PCa. Of these, 57 491 men (36%) underwent at least one PSA test. Older age group was strongly related to a greater number of tests over 10 years (p<0.001, table 2).

### PSA levels and retesting
Data on PSA levels in CPRD were incomplete, but the median first PSA result of those tested with a result (n=119 175) was 1.23 ng/mL (IQR 0.70–2.60; figure 2). If those with a PSA test date but missing level (n=1522, 1%) had a result of approximately 0, the median PSA would be 1.20 ng/mL (IQR 0.70–2.60). Removing the lowest age category (45–49) increased the median PSA to 1.34 ng/mL (0.70–2.90), n=102 107. For the ProtecT men, the median PSA result of those collected (n=58 542) was 0.99 ng/mL (IQR 0.60–1.70).

Of those PSA-tested with a full year's follow-up after their test (aged 50–69 years), 17 757/90 252 (20%) had a second test within a year of their first; of which 17 218 had an initial result (table 3). Undergoing a second PSA test within a year of the first test was strongly associated with a higher PSA level at the first test (OR per PSA category higher 1.85, 95% CI 1.83 to 1.88; p<0.001). Those men with a PSA <3 ng/mL were more likely to be retested

within a year if they were in an older age group (OR per age category older 1.04, 95% CI 1.04 to 1.04; p<0.001). This trend was reversed for those men with a PSA ≥3 ng/mL, those in an older age group were less likely to be retested within a year than those in a younger age group (OR per age category older 0.98, 95% CI 0.97 to 0.98; p<0.001).

Older men were also at greater risk of having a higher PSA test than those in younger age categories, for ProtecT and CPRD data (for the CPRD cohort, OR per age group older1.08, 95% CI 1.08 to 1.08; p<0.001). On average, those ProtecT participants presenting with LUTS appeared to have higher PSA levels than those with no LUTS, whereas men in CPRD had the highest PSA results (table 3).

### Subsequent biopsies and diagnoses
From the ProtecT data, 22 200 men were identified as having LUTS (based on our definition) at the consultation for their PSA test, and 36 364 men did not have LUTS; 22 193 and 36 349, respectively, had a PSA result recorded. For men with a PSA level of 3 ng/mL or higher, biopsy and diagnosis rates were much higher in ProtecT participants than CPRD. This remained true, even when those with a high PSA level were confirmed high in a further test (online supplementary table S2); furthermore, for men in the CPRD cohort, a lower proportion underwent biopsy than were subsequently diagnosed. Overall, the odds of diagnosis within a year of a PSA test

**Table 1** Factors that influence the risk of having a PSA test/prostate cancer diagnosis

| | N (%) | PSA testing | | | PCa diagnosis | | |
| --- | --- | --- | --- | --- | --- | --- | --- |
| | | Men who had at least one PSA test | 10-year risk % (95% CI)* | p Value | Men who had a PCa diagnosis | 10-year risk % (95% CI)* | p Value |
| All men | 434 236 (100%) | 120 697 | 39.19 (39.01 to 39.38) | | 7538 | 2.72 (2.66 to 2.78) | |
| **Age (in 2002)** | | | | | | | |
| 45–49 | 104 782 (24%) | 17 297 | 25.20 (24.86 to 25.55) | | 296 | 0.49 (0.44 to 0.55) | |
| 50–54 | 100 211 (23%) | 24 162 | 34.70 (34.33 to 35.08) | | 858 | 1.40 (1.31 to 1.50) | |
| 55–59 | 97 224 (22%) | 30 328 | 43.08 (42.69 to 43.47) | p<0.001† | 1700 | 2.76 (2.63 to 2.90) | p<0.001† |
| 60–64 | 71 637 (17%) | 25 518 | 48.57 (48.11 to 49.04) | | 2179 | 4.67 (4.47 to 4.87) | |
| 65–69 | 60 381 (14%) | 23 392 | 52.95 (52.44 to 53.45) | | 2505 | 6.28 (6.04 to 6.53) | |
| **Region** | | | | | | | |
| South East Coast | 51 494 (12%) | 17 434 | 47.45 (46.90 to 48.01) | | 998 | 3.14 (2.95 to 3.34) | |
| Wales | 35 277 (8%) | 12 119 | 45.02 (44.40 to 45.66) | | 689 | 2.79 (2.59 to 3.01) | |
| Northern Ireland | 12 730 (3%) | 4515 | 43.69 (42.70 to 44.69) | | 264 | 2.75 (2.44 to 3.11) | |
| South Central | 53 577 (12%) | 16 383 | 42.45 (41.93 to 42.98) | | 976 | 2.79 (2.62 to 2.98) | |
| South West | 44 060 (10%) | 12 399 | 40.82 (40.22 to 41.42) | | 777 | 2.96 (2.75 to 3.18) | |
| West Midlands | 40 677 (9%) | 11 453 | 39.28 (38.69 to 39.88) | p<0.001‡ | 704 | 2.66 (2.47 to 2.86) | p<0.001‡ |
| North West | 56 484 (13%) | 16 340 | 38.88 (38.39 to 39.37) | | 994 | 2.54 (2.38 to 2.70) | |
| East of England | 47 851 (11%) | 12 386 | 38.85 (38.26 to 39.44) | | 810 | 2.88 (2.68 to 3.09) | |
| Yorkshire and The Humber | 18 717 (4%) | 4131 | 35.49 (34.51 to 36.50) | | 251 | 2.40 (2.10 to 2.75) | |
| East Midlands | 19 539 (5%) | 4466 | 34.43 (33.50 to 35.38) | | 260 | 2.40 (2.10 to 2.75) | |
| North East | 8113 (2%) | 1859 | 30.49 (29.31 to 31.71) | | 123 | 2.25 (1.88 to 2.69) | |
| Scotland | 45 717 (11%) | 7212 | 23.82 (23.31 to 24.33) | | 692 | 2.39 (2.21 to 2.58) | |
| **IMD (quartiles)** | | | | | | | |
| 1–5 (least deprived) | 84 706 (20%) | 29 422 | 46.26 (45.84 to 46.67) | | 1824 | 3.20 (3.06 to 3.36) | |
| 6–10 | 69 496 (16%) | 21 611 | 42.45 (42.00 to 42.91) | | 1332 | 2.92 (2.76 to 3.08) | |
| 11–15 | 56 865 (13%) | 14 596 | 36.32 (35.82 to 36.82) | p<0.001† | 916 | 2.54 (2.38 to 2.72) | p<0.001† |
| 16–20 (most deprived) | 40 833 (9%) | 8735 | 31.92 (31.33 to 32.51) | | 483 | 1.92 (1.75 to 2.10) | |
| No IMD recorded | 182 336 (42%) | 46 333 | 36.87 (36.58 to 37.16) | | 2983 | 2.63 (2.53 to 2.73) | |
| **Pre-registration PSA test** | | | | | | | |
| Previously tested | 27 211 (6%) | 15 368 | 73.21 (72.54 to 73.89) | p<0.001‡ | 1089 | 5.94 (5.59 to 6.31) | p<0.001‡ |
| Not previously tested | 407 025 (94%) | 105 329 | 36.97 (36.78 to 37.16) | | 6449 | 2.51 (2.45 to 2.57) | |

*Kaplan-Meier failure function at 10 years.
†p for trend.
‡p across categories.
IMD, index of multiple deprivation; PCa, prostate cancer; PSA, prostate-specific antigen.

**Table 2** Number of PSA tests* received by men with full 10-year follow-up and no prostate cancer diagnosis

| Number of tests | 0 | 1 | 2 | 3 | 4 | 5 | 6 | 7 | 8 | 9 | ≥10 |
|---|---|---|---|---|---|---|---|---|---|---|---|
| All men | 100095 (64%) | 28561 (18%) | 12196 (8%) | 6047 (4%) | 3449 (2%) | 2050 (1%) | 1379 (1%) | 929 (1%) | 734 (<1%) | 539 (<1%) | 1607 (1%) |
| Age at entry in 2002 | | | | | | | | | | | |
| 45–49 | 28998 (77%) | 5651 (15%) | 1744 (5%) | 647 (2%) | 284 (1%) | 162 (<1%) | 81 (<1%) | 37 (<1%) | 23 (<1%) | 20 (<1%) | 47 (<1%) |
| 50–54 | 25299 (68%) | 6846 (18%) | 2549 (7%) | 1132 (3%) | 606 (2%) | 324 (1%) | 195 (1%) | 107 (<1%) | 100 (<1%) | 63 (<1%) | 150 (<1%) |
| 55–59 | 21471 (59%) | 7092 (20%) | 3176 (9%) | 1654 (5%) | 866 (2%) | 549 (2%) | 384 (1%) | 246 (1%) | 195 (1%) | 144 (<1%) | 380 (1%) |
| 60–64 | 13903 (54%) | 4941 (19%) | 2508 (10%) | 1371 (5%) | 917 (4%) | 524 (2%) | 354 (1%) | 277 (1%) | 212 (1%) | 160 (1%) | 482 (2%) |
| 65–69 | 10424 (50%) | 4031 (19%) | 2219 (11%) | 1243 (6%) | 776 (4%) | 491 (2%) | 365 (2%) | 262 (1%) | 204 (1%) | 152 (1%) | 548 (3%) |

*PSA tests without a level recorded were included in these totals.
PSA, prostate-specific antigen.

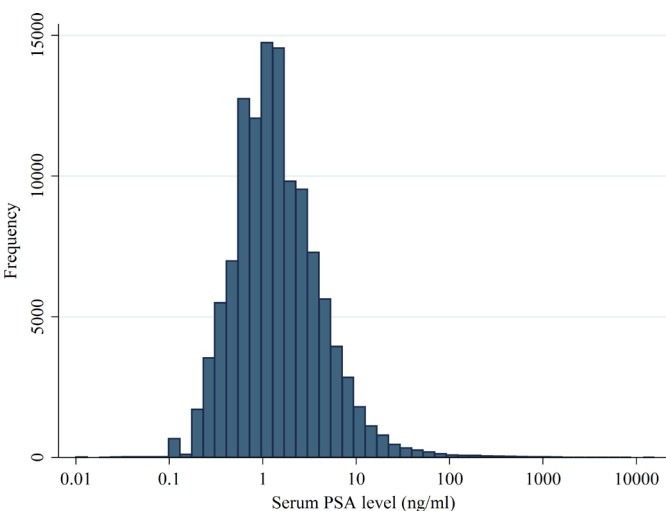

**Figure 2** Distribution of PSA levels on the log scale. PSA, prostate-specific antigen.

was three times higher in the ProtecT study compared with the CPRD data for those with PSA ≥3 (OR 2.99, 95% CI 2.80 to 3.18; p<0.001).

For CPRD, as expected, men with a higher PSA level were more likely to be diagnosed (OR per PSA category higher 3.52, 95% CI 3.42 to 3.61; p<0.001), as were older men (OR per age category higher 1.08, 95% CI 1.07 to 1.08; p<0.001). For those aged between 50 and 69 years, the biopsy rates were <1%, 1%, 5%, 10% and 13% for PSA categories PSA <3, 3≤PSA<4, 4≤PSA<6, 6≤PSA<10 and PSA ≥10, respectively. The diagnosis rates were <1%, 1%, 6%, 18% and 45%, respectively.

## DISCUSSION

This paper has examined the risk of receiving a PSA test over a 10-year period in a large retrospective cohort of men aged 45–69 years in the UK (excluding London). The 10-year risk of undergoing a PSA test was estimated at 39.2%, whereas the 10-year risk of receiving a PCa diagnosis was estimated at 2.7%. Higher rates of both testing and diagnoses were associated with older age, more southerly region of residence, less deprived IMD and a history of PSA testing. For all age groups and PSA levels, the proportion of men undergoing biopsy and subsequently diagnosed with PCa following a PSA test in UK GP is low when compared with men in the PSA testing programme undertaken as part of the ProtecT trial.[17]

Overall, the number of men without a prior diagnosis of PCa receiving at least one PSA test over 10 years is high, especially given the lack of a screening programme in the UK. The higher rates of testing in older men are consistent with the findings of other studies of UK GP[8–10] although not with the age distribution of men agreeing to participate in the ProtecT study, the latter being in close agreement with the male population age distribution, with the majority of men being younger than 60 years.[17] These findings suggest that interest in PCa screening is not concentrated in the older age groups, and that the

**Table 3** PSA levels* by age group in the Clinical Practice Research Datalink (CPRD) and Prostate Testing for Cancer and Treatment (ProtecT) study men (Lower Urinary Tract Symptoms (LUTS) vs no LUTS)

| PSA level | Age group 50–54 | | | Age group 55–59 | | | Age group 60–64 | | | Age group 65–69 | | |
|---|---|---|---|---|---|---|---|---|---|---|---|---|
| | CPRD† | LUTS‡ | No LUTS‡ | CPRD† | LUTS‡ | No LUTS‡ | CPRD† | LUTS‡ | No LUTS‡ | CPRD† | LUTS‡ | No LUTS‡ |
| **Number of men tested (%)** | | | | | | | | | | | | |
| PSA <3 | 17632 (87%) | 5270 (95%) | 10261 (96%) | 20886 (80%) | 5724 (90%) | 10321 (92%) | 16010 (72%) | 4715 (83%) | 7534 (86%) | 13338 (66%) | 3611 (78%) | 4633 (82%) |
| 3≤PSA<4 | 936 (5%) | 138 (2%) | 217 (2%) | 1714 (7%) | 280 (4%) | 418 (4%) | 1734 (8%) | 411 (7%) | 461 (5%) | 1776 (9%) | 390 (8%) | 392 (7%) |
| 4≤PSA<6 | 838 (4%) | 79 (1%) | 138 (1%) | 1655 (6%) | 202 (3%) | 294 (3%) | 1868 (8%) | 266 (5%) | 404 (5%) | 2026 (10%) | 326 (7%) | 338 (6%) |
| 6≤PSA<10 | 500 (2%) | 29 (1%) | 61 (1%) | 1092 (4%) | 101 (2%) | 146 (1%) | 1340 (6%) | 172 (3%) | 194 (2%) | 1642 (8%) | 200 (4%) | 185 (3%) |
| 10≤PSA<20 | 294 (1%) | 14 (<1%) | 16 (<1%) | 541 (2%) | 29 (<1%) | 42 (<1%) | 717 (3%) | 64 (1%) | 97 (1%) | 925 (5%) | 88 (2%) | 75 (1%) |
| PSA ≥20 | 168 (1%) | 3 (<1%) | 7 (<1%) | 350 (1%) | 16 (<1%) | 21 (<1%) | 513 (2%) | 28 (1%) | 38 (<1%) | 614 (3%) | 37 (1%) | 52 (1%) |
| **Number retested within 1 year of their first PSA test (%)** | | | | | | | | | | | | |
| PSA <3 | 1580 (9%) | | | 2263 (11%) | | | 2148 (13%) | | | 1992 (15%) | | |
| 3≤PSA<4 | 256 (27%) | | | 336 (20%) | | | 308 (18%) | | | 359 (20%) | | |
| 4≤PSA<6 | 507 (61%) | | | 902 (55%) | | | 927 (50%) | | | 819 (40%) | | |
| 6≤PSA<10 | 336 (67%) | | | 681 (62%) | | | 778 (58%) | | | 934 (57%) | | |
| 10≤PSA<20 | 185 (63%) | | | 315 (58%) | | | 408 (57%) | | | 489 (53%) | | |
| PSA ≥20 | 82 (49%) | | | 163 (47%) | | | 215 (42%) | | | 235 (38%) | | |
| **Number biopsied within 1 year of their first PSA test—with or without biopsy (%)** | | | | | | | | | | | | |
| PSA <3 | 16 (<1%) | 0 (0%) | 0 (0%) | 33 (<1%) | 0 (0%) | 0 (0%) | 21 (<1%) | 0 (0%) | 0 (0%) | 22 (<1%) | 0 (0%) | 0 (0%) |
| 3≤PSA<4 | 13 (1%) | 122 (88%) | 174 (80%) | 7 (<1%) | 245 (88%) | 347 (83%) | 10 (1%) | 345 (84%) | 374 (81%) | 5 (<1%) | 316 (81%) | 307 (78%) |
| 4≤PSA<6 | 61 (7%) | 74 (94%) | 122 (88%) | 92 (6%) | 186 (92%) | 254 (86%) | 87 (5%) | 240 (90%) | 339 (84%) | 50 (2%) | 278 (85%) | 274 (81%) |
| 6≤PSA<10 | 51 (10%) | 27 (93%) | 56 (92%) | 133 (12%) | 92 (91%) | 134 (92%) | 128 (10%) | 160 (93%) | 166 (86%) | 139 (8%) | 168 (84%) | 161 (87%) |
| 10≤PSA<20 | 36 (12%) | 13 (93%) | 16 (100%) | 84 (16%) | 26 (90%) | 41 (98%) | 109 (15%) | 60 (94%) | 89 (92%) | 116 (13%) | 81 (92%) | 65 (87%) |
| PSA ≥20 | 18 (11%) | 3 (100%) | 5 (71%) | 56 (16%) | 11 (69%) | 17 (81%) | 62 (12%) | 23 (82%) | 33 (87%) | 66 (11%) | 30 (81%) | 39 (75%) |
| **Number diagnosed within 1 year of their first PSA test (%)** | | | | | | | | | | | | |
| PSA <3 | 16 (<1%) | 0 (0%) | 2 (<1%) | 46 (<1%) | 1 (<1%) | 2 (<1%) | 48 (<1%) | 2 (<1%) | 1 (<1%) | 48 (<1%) | 2 (<1%) | 1 (<1%) |
| 3≤PSA<4 | 19 (2%) | 28 (20%) | 47 (22%) | 25 (1%) | 62 (22%) | 102 (24%) | 18 (1%) | 98 (24%) | 115 (25%) | 10 (1%) | 87 (22%) | 111 (28%) |
| 4≤PSA<6 | 80 (10%) | 29 (37%) | 44 (32%) | 124 (7%) | 64 (32%) | 95 (32%) | 129 (7%) | 80 (30%) | 139 (34%) | 79 (4%) | 90 (28%) | 119 (35%) |
| 6≤PSA<10 | 99 (20%) | 15 (52%) | 27 (44%) | 186 (17%) | 37 (37%) | 66 (45%) | 258 (19%) | 70 (41%) | 96 (49%) | 264 (16%) | 74 (37%) | 84 (45%) |
| 10≤PSA<20 | 87 (30%) | 6 (43%) | 12 (75%) | 155 (29%) | 18 (62%) | 29 (69%) | 242 (34%) | 35 (55%) | 76 (78%) | 296 (32%) | 49 (56%) | 47 (63%) |
| PSA ≥20 | 86 (51%) | 3 (100%) | 7 (100%) | 220 (63%) | 11 (69%) | 16 (76%) | 347 (68%) | 23 (82%) | 34 (89%) | 421 (69%) | 32 (86%) | 42 (81%) |

*Those without a test date could not be included as we could not determine whether they had a full year's follow-up post-test.
†Data taken between January 2002 and December 2011 for PSA tests taken from January 2002 to December 2010—any men without 1 full year's follow-up post-test were removed.
‡Data taken from the ProtecT study[17] between January 2002 and January 2010 for PSA tests taken from January 2002 to January 2009.
CPRD, Clinical Practice Research Datalink; LUTS, lower urinary tract symptoms; PSA, prostate-specific antigen.

greater incidence of testing in older men is likely to arise due to other diagnostic indications for the PSA test. The increase in testing with age could be due to an increase in LUTS with age[22] or with the GP wanting to rule out the possibility of PCa[23] despite this rarely being the cause of such symptoms.[21 24] It is also thought that the PSA level is a useful indicator of prostate volume and may inform the choice between treatment options for benign prostatic hyperplasia and other benign conditions.[12 25]

The observation of greater testing of men living in more affluent areas is consistent with previous studies.[8–10] This association presumably arises from more affluent men being more likely to request a test or through GPs serving more affluent areas being more likely to promote the test to their male patients. There is some evidence to suggest that PCa is more prevalent in areas of lower deprivation[26]; however, the extent to which PSA testing patterns inform this is difficult to determine.

Overall, 11% of men with 10 years of follow-up were tested three or more times. This varied by age group, 20% of men aged 65 to 69 years at the outset being tested three or more times, compared with 3% of men aged 45 to 49 years. Two major trials of PCa screening have employed repeated PSA testing,[27 28] although the association with age would not be expected if the programmes in those trials were being followed by UK general practitioners, and certainly not a greater number of older men undergoing multiple tests.

ProtecT participants with LUTS had slightly higher PSA levels on average than those without LUTS. While 78% of men in UK GP undergoing a PSA test were found to have a level below 3 ng/mL, on average, PSA levels were higher than seen in the ProtecT study. In part, this will be due to the older age profile of men in UK GP compared with ProtecT participants, but it is also consistent with more tests in GP being undertaken to inform a diagnosis of LUTS, as LUTS are associated with elevated PSA levels (there is no reason to suppose a higher prevalence of non-symptomatic and undiagnosed PCa in CPRD or ProtecT GPs). A strong association was observed for all age groups between higher PSA levels at a first test and the probability of a man undergoing a second test within 1 year, indicating that the results of the PSA tests did inform clinical management.

The incidence of biopsy and PCa diagnosis in the CPRD cohort suggests that a PSA of 4 ng/mL or more was being used in UK GP as a trigger for further diagnostic investigations. The incidence of biopsy in the CPRD cohort is very low, and the fact that there are fewer biopsies than PCa diagnoses suggests that either many more men were refusing biopsy, perhaps due to the full screening process not being discussed at the time of PSA testing, or that biopsy is being under-recorded in GP data. However, even allowing for a degree of under-reporting, only a small minority of men with high PSA levels were recorded as having had a biopsy, which contrasts with 80% plus of ProtecT men with PSA of 3 ng/mL or higher undergoing the investigation.

Furthermore, the risk of a PCa diagnosis in the CPRD cohort is much lower than in comparable men participating in the ProtecT study prospective PSA testing programme. These findings are again consistent with the majority of PSA tests in UK GP being undertaken to inform the diagnosis and management of LUTS in older men, with no intention of screening for PCa.

## STRENGTHS AND LIMITATIONS

The major strength of this investigation is the use of CPRD data which has allowed a large retrospective cohort of men to be constructed and followed up for a period of up to 10 years. The use of CPRD data is also behind the key weakness of this study: the completeness of some data items is uncertain, with the recorded diagnoses outnumbering the recorded biopsies indicating that the latter are under-recorded, presumably even when cancer is diagnosed.

We did not attempt to distinguish those tests undertaken in men presenting with and without symptoms, and this could be considered a further limitation of our study. Screening aims to diagnose a disease before symptoms arise. However, PCa rarely results in LUTS and sexual symptoms until it is at an advanced stage. For the vast majority of men with urinary and sexual symptoms, the cause is benign, and in fact, men with an elevated PSA are less likely to be diagnosed with PCa if they also have LUTS or impaired sexual function.[21 29] The pattern of PSA testing in the CPRD cohort suggests that many PSA tests are being undertaken to inform the diagnosis and management of LUTS and knowing which men had been PSA-tested because of a presentation with symptoms would have lent further support to this hypothesis.

## CONCLUSION

In UK GP, 39.2% of men aged 45 to 69 years and initially free of PCa undergo at least one PSA test during a 10-year follow-up period (2002 to 2011). However, testing rates are higher in the older age groups, and high PSA levels are commonly not followed up by a biopsy, required for the diagnosis of PCa. Hence, it is likely that a high proportion of these tests are related to investigations or management of LUTS and other benign conditions and cannot be considered as part of an effective (informal) effort to screen for PCa.

**Acknowledgements** We acknowledge the contribution of the CAP trial group. Investigators: RM (lead principal investigator (PI)), JD (PI), DN (PI), FH (PI), ET (trial coordinator), CM (statistician), Jonathan Sterne (statistician), Sian Noble (health economist) and AL. Research staff: Elizabeth Hill, Siaw Yien Ng, Naomi Williams, Liz Down (data manager), EW (data manager), GY (statistician), Joanna Thorn (health economist), Charlotte Davies, Laura Hughes, Mari-Anne Rowlands and Lindsey Bell. Management committee: ET (Chair), RM, JD, CM, Jonathan Sterne, Sian Noble, YB-S, AL, SO, Peter Brindle and SE. Trial steering committee: Michael Baum (Chair), Peter Albertsen, Tracy Roberts, Mary Robinson, Jan Adolfsson, David Dearnaley, AZ, Fritz Schröder, Tim Peters, Peter Holding, Teresa Lennon, Sue Bonnington, Malcolm Mason, Jon Oxley, RM, JD, DN, FH, ET and AL. Data monitoring committee: Lars Holmberg (Chair), Robert Pickard, Simon Thompson and Usha Menon. Cause of death committee: Peter Albertsen (Chair), Colette Reid, John McFarlane, Jon Oxley,

Mary Robinson, Jan Adolfsson, Michael Baum, Anthony Zietman, Amit Bahl and Anthony Koupparis. Administrative staff: Marta Tazewell and Genevieve Hatton-Brown. We wish to extend our thanks to Pete Shiarly for the development of bespoke databases. We also wish to acknowledge the contribution of all members of the ProtecT study research groups. Thanks are extended to the Cancer Registries and staff at the Health and Social Care Information Centre. We acknowledge the contributions of the ProtecT and CAP study participants, investigators, researchers, data monitoring committees and trial steering committees.

**Contributors** ELT, SEO, YB-S, JAL, DEN, FCH, JLD, RMM and CM contributed to the design of the study and formulated the research question. SH, ELT, EIW, RMM and CM contributed to the acquisition of the data. GJY, SH, ELT, EIW, SE, RMM and CM made substantial contributions to the analysis and interpretation of data for the work. GJY, SH and CM wrote the first draft. All authors commented on the drafts and revised it critically for important intellectual content. All authors approved the final manuscript and are accountable for the accuracy and integrity of the work.

**Funding** This work was supported by the Cancer Research UK and the UK Department of Health for the CAP trial (C11043/A4286, C18281/A8145, C18281/A11326 and C18281/A15064). CAP is sponsored by the University of Bristol and is registered at Current Controlled Trials (ISRCTN92187251). The ProtecT trial is funded by the UK National Institute for Health Research (NIHR), Health Technology Assessment Programme (projects 96/20/06 and 96/20/99). The views and opinions expressed herein are our own and do not necessarily reflect those of the Department of Health. The ProtecT trial is sponsored by the University of Oxford and is registered at Current Controlled Trials (ISRCTN20141297) and ClinicalTrials.gov (NCT02044172). We acknowledge the support from the Oxford NIHR Biomedical Research Centre through the Surgical Innovation and Evaluation Theme and the Surgical Interventional Trials Unit and Cancer Research UK through the Oxford Cancer Research Centre. SH is a Wellcome Trust-funded PhD student with grant code 102432/Z/13/Z. JD is supported, in part, by the NIHR Collaboration for Leadership in Applied Health Research and Care West, hosted by the University Hospitals Bristol NHS Foundation Trust. FH is supported, in part, by the Oxford NIHR Biomedical Research Centre and the Cancer Research UK Oxford Centre. RM is supported, in part, by the University Hospitals Bristol NHS Foundation Trust National Institute for Health Research Bristol Nutrition Biomedical Research Unit and CRUK (C18281/A19169). JD, FH and DN are NIHR senior investigators. The views and opinions expressed herein are the authors own and do not necessarily reflect those of the Department of Health. The funders and sponsor had no role in the design and conduct of the study, the preparation of the report or the decision to publish. The corresponding author had full access to the data and takes responsibility for the decision to submit for publication.

**Competing interests** None declared.

**Patient consent** Detail has been removed from this case description/these case descriptions to ensure anonymity. The editors and reviewers have seen the detailed information available and are satisfied that the information backs up the case the authors are making.

**Ethics approval** ProtecT study: East Midlands Multicentre Research Ethics Committee (01/4/025).

**Provenance and peer review** Not commissioned; externally peer reviewed.

**Data sharing statement** The authors do not have permission to share data obtained from the Clinical Practice Research Datalink (CPRD). Baseline data from ProtecT may be available on application (email info-protect@bris.ac.uk).

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
