## [Reviewer comments · BMJ Open]

ARTICLE DETAILS

TITLE (PROVISIONAL)	Prostate Specific Antigen (PSA) testing of men in UK general practice: a 10-year longitudinal cohort study
AUTHORS	Young, Grace; Harrison, Sean; Turner, Emma; Walsh, Eleanor; Oliver, Steven; Ben-Shlomo, Yoav; Evans, Simon; Lane, Athene; Neal, David; Hamdy, Freddie; Donovan, Jenny; Martin, Richard; Metcalfe, Chris

VERSION 1 – REVIEW

REVIEWER	Tobias Nordström Karolinska Institutet, Sweden
REVIEW RETURNED	26-May-2017

GENERAL COMMENTS	Thank you for the opportunity to review this well written manuscript using large-scale, patient-level data together with ProtecT data to describe the use of PSA and prostate biopsies in the UK. The authors report moderately intense PSA testing behaviour and very low prostate biopsy rates. To the best of my understanding, the methodology is sound and the reporting of results and conclusions valid. However, I have a few concerns that might be addressed in greater detail to enhance the understanding of the manuscript. (i) As the authors state, there is a risk of missing data/ under-reporting of PSA levels and number of biopsy procedures. While this is central in manuscript interpretation, please give a description in the methodology section on how missing data was handled. Some kind of sensitivity analysis on how prostate biopsy underreporting affect the results would enhance the possible conclusions. (ii) Table 2 gives the number of PSA test during 10 years by age. It could be clarified that the age given is the age at entry (it is, right?) (iii) The authors state that "Overall the number of men without a prior diagnosis of prostate cancer receiving at least one PSA test over 10-years is high, especially given the lack of a screening programme ...". Is it really, given the evidence for a mortality reduction using PSA screening and the shared-decision strategy to perform testing? (iv) I was confused by the statement that "It is also thought that the PSA level is a useful indicator of prostate volume and may inform the choice between treatment options for BPH and other benign conditions." Do the authors argue that PSA might be used solely for benign diagnostics of BPH, without considering prostate cancer risk?
---

	(v) The authors describe dramatically low prostate biopsy rates, and give the possible explanation that GPs use PSA for BPH diagnostics only. A clearer discussion on the pros and cons of this clinical practice might be justified to increase value for enhancing clinical practice.
--	---

REVIEWER	Rebecka Arnsrud Godtman Department of Urology, Institute of clinical sciences, Sahlgrenska Academy at the University of Gothenburg, Sweden
REVIEW RETURNED	12-Jun-2017

GENERAL COMMENTS	This is an interesting study which aims at answering an important question. However, there are some major concerns which should be addressed:  -Could the authors give some information regarding the completeness of the data in the CPRD. There seem to be severe underreporting of the proportion of men having a prostate biopsy. The fact that only 6% were registered as having a prostate biopsy after a raised PSA but 15% had received a prostate cancer diagnose shows that the data on prostate biopsies cannot be used. The only conclusion that can be drawn regarding prostate biopsies is that they are severely underreported in the CPRD. Please remove data on prostate biopsies in table 3 and change the text in the abstract, result and discussion. If the authors want to investigate whether the PSA tests should be considered as screening tests leading to further investigations one alternative could perhaps be to use the same strategy as Moss et al (ref#10) and instead look at the rates of referral. -Including information on symptoms would improve the paper. According to the Moss et al. (ref #10) paper this information can be found in the CPRD. -Has this study been approved by an Ethical committee? Please add this information. -Please explain why the (674-578=96) 96 practices were excluded. Were these the London practices or were these the practices which did not reach acceptable research standard? -I strongly disagree with the assumption that those with missing data on PSA level are recorded as undetectable (PSA=0). I find it highly unlikely that they have PSA level zero. -Subjects and methods: Please define "acceptable research standard" -Please consider performing competing risk analyses with cumulative incidence function instead of Kaplan Meier analyses. The age groups have different risks for competing risks, i.e. risk of prostate cancer and/or death. -In the analyses regarding retesting, I find it strange to have PSA-levels stratified in groups in the model and the groups do not have the same range (PSA<3, 3-4, 4-6, 6-10 etc). Why is not PSA and age used as continuous variables? -According to the Discussion section "The proportion of men undergoing their first test in the follow-up period increases steadily throughout the ten-year period" but I cannot find anywhere in the result section that calendar year has been explored. -Consider changing some of the references. Refer to the original source. For example, refer to the specific guidelines and to an article which investigates PSA-testing patterns around the world and not Harvey et al. (ref #1) which is an article about TRUS and Prensner et al. (ref #2) which is about the next generation of biomarkers.
--

VERSION 1 – AUTHOR RESPONSE

Reviewer: 1

Comment 1. As the authors state, there is a risk of missing data/ under-reporting of PSA levels and number of biopsy procedures. While this is central in manuscript interpretation, please give a description in the methodology section on how missing data was handled. Some kind of sensitivity analysis on how prostate biopsy underreporting affect the results would enhance the possible conclusions.

Response: Our primary aim for this study was to investigate patterns of PSA testing in UK primary care, of men with up to 10 years follow-up. We obtained routine primary care records to investigate this; of course, there may be a degree of under-reporting of PSA tests in such routine data, although the annual testing rates we observe are comparable to other studies using other sources of data. Biopsy rates are very likely to be under-reported in our data, although Moss et al (2016)* also find low rates when examining records of referral. Whilst we wish to present the biopsy data to inform other researchers in this area (as do Moss et al*), we can interpret our PSA testing data equally well with information on diagnoses. Data on diagnoses are going to be near complete as they are collected through a link to the cancer registries. The likely under-reporting of biopsy is not informing our conclusions, so we have not conducted the suggested sensitivity analyses.

Comment 2. Table 2 gives the number of PSA test during 10 years by age. It could be clarified that the age given is the age at entry (it is, right?)

Response: The reviewer is right; we have attempted to clarify this by changing the column heading: "Age at entry in 2002".

Comment 3. The authors state that "Overall the number of men without a prior diagnosis of prostate cancer receiving at least one PSA test over 10-years is high, especially given the lack of a screening programme ..." Is it really, given the evidence for a mortality reduction using PSA screening and the shared-decision strategy to perform testing?

Response: The authors feel that it is high given the current recommendations adopted in the UK at this present time. Melia et al (2004)** have previously commented on the proportions of men attending a PSA test by asymptomatic (33%), symptomatic (47%) and retesting (20%). Therefore, of the ~40% of men tested, we expect that the screening (asymptomatic men) is closer to ~13%. While there is evidence from the ERSPC that PSA testing is effective in reducing mortality, these results were not published until 2009 and we only collected data until 2012. The NHS website currently states: "There is currently no screening programme for prostate cancer in the UK. This is because it has not been proved that the benefits would outweigh the risks."

Comment 4. I was confused by the statement that "It is also thought that the PSA level is a useful indicator of prostate volume and may inform the choice between treatment options for BPH and other benign conditions." Do the authors argue that PSA might be used solely for benign diagnostics of BPH, without considering prostate cancer risk?

Response: Our sources suggest that the PSA level can be considered as a useful predictor of BPH. Roehrborn et al. (1999)*** felt that the PSA may predict those men at increased risk of developing acute urinary retention or needing BPH-related surgery. NICE guidelines (2010)**** also state that the cut off level of 1.4 may direct drug treatment decision for those with bladder outlet obstruction or benign prostatic enlargement.

With this statement, we are suggesting that a PSA test may be prompted by the management of what is thought to be BPH, although we expect that prostate cancer risk would subsequently be considered if the PSA level was found to be high.

Comment 5. The authors describe dramatically low prostate biopsy rates, and give the possible explanation that GPs use PSA for BPH diagnostics only. A clearer discussion on the pros and cons of this clinical practice might be justified to increase value for enhancing clinical practice.

Response: While the authors do find this interesting it goes beyond the scope of this paper. This paper aims to understand routine data on practices in UK primary care regarding PSA testing for prostate cancer. It may be of interest to consider this in a future paper.

Reviewer: 2

Comment 1. Could the authors give some information regarding the completeness of the data in the CPRD. There seem to be severe underreporting of the proportion of men having a prostate biopsy. The fact that only 6% were registered as having a prostate biopsy after a raised PSA but 15% had received a prostate cancer diagnose shows that the data on prostate biopsies cannot be used. The only conclusion that can be drawn regarding prostate biopsies is that they are severely underreported in the CPRD. Please remove data on prostate biopsies in table 3 and change the text in the abstract, result and discussion. If the authors want to investigate whether the PSA tests should be considered as screening tests leading to further investigations one alternative could perhaps be to use the same strategy as Moss et al (ref#10) and instead look at the rates of referral.

Response: With regards to the biopsy data, we would prefer to present available data on biopsies as readers of this manuscript will expect to see them. Our interpretation of the biopsy data is entirely consistent with both yourself and the additional reviewer. We consider the biopsy and diagnosis rates together to avoid drawing misleading conclusions. Data on diagnoses are going to be near complete as they are collected through a link to the cancer registries. With regards to rates of referral, Moss et al (2016)* believe that these are also under-recorded. We believed (wrongly) that the results of a procedure would be recorded at least as well as referrals and so we only requested data on the procedure itself from CPRD, we do not have the data on referral.

Comment 2. Including information on symptoms would improve the paper. According to the Moss et al. (ref #10) paper this information can be found in the CPRD.

Response: This is an interesting idea but sadly we did not request information on symptoms from CPRD. Our work with the ProtecT data indicates that LUTS are not prognostic of diagnosis following a PSA test. We do discuss this in the discussion and present diagnostic process for ProtecT men with and without LUTS. LUTS were measured routinely and with validated questionnaires in the ProtecT study.

Comment 3. Has this study been approved by an Ethical committee? Please add this information.

Response: The following statement has now been added to the methods section (page 8): The CPRD group holds ethical approval from a National Research Ethics Service Committee (NRECS) for all purely observational research using anonymised CPRD data. The ProtecT trial holds ethics approval from the Trent Multicentre Research Ethics Committee (Trent MREC), 21/06/2001, ref: 01/4/025.

Comment 4. Please explain why the (674-578=96) 96 practices were excluded. Were these the London practices or were these the practices which did not reach acceptable research standard?

Response: These practices are a combination of the two and were removed by CPRD themselves. Therefore, it would be difficult to differentiate between those that were excluded because they were in London and those who did not reach “acceptable research standard”.

Comment 5: I strongly disagree with the assumption that those with missing data on PSA level are recorded as undetectable (PSA=0). I find it highly unlikely that they have PSA level zero.

Response: Apologies for any misunderstanding, when we say, “equivalent to zero” we mean below the detection threshold and therefore less than 0.1. We have now added this clarification to the manuscript (page 7). We stated the median and interquartile range for the PSA levels, with and without this assumption and there was little difference: 1.23ng/ml vs. 1.20ng/ml. Had it made a larger impact then we may have considered presenting results with and without this assumption.

Comment 6: Subjects and methods: Please define “acceptable research standard”

Response: This is all in-house at CPRD and we therefore can't be sure of the definition. More details can be found in the supplementary material of a paper by Herrett et al (2015)^{*****}. It is based on acceptability of patient data (registration status, recording of events, valid age, valid gender) and practice data (gaps in reporting of treatment/deaths). They go on to explain that these criteria do not ensure data quality. We have added the reference to the text.

Comment 7: Please consider performing competing risk analyses with cumulative incidence function instead of Kaplan Meier analyses. The age groups have different risks for competing risks, i.e. risk of prostate cancer and/or death.

Response: Our primary interest is in the occurrence of PSA testing. We agree that older men are both more likely to be tested, and to die of any cause. However, we are unaware of other groups presenting cumulative incidence functions, and the age-specific estimates we present (Table 1, and Figure 1) will avoid overestimation of PSA testing rates due to the competing risk of death as men get older. In the recent analysis of the ProtecT primary outcome, prostate cancer mortality, a sensitivity analysis that accommodated any competing risk from all-cause mortality gave identical results to the standard proportional hazards estimates of treatment effects (Hamdy et al. 2016^{*****})

Comment 8: In the analyses regarding retesting, I find it strange to have PSA-levels stratified in groups in the model and the groups do not have the same range (PSA<3, 3-4, 4-6, 6-10 etc). Why is not PSA and age used as continuous variables?

Response: We used wider categories to ensure adequate numbers per group (added to page 8). We have, as you suggested, computed the p values using continuous measures of age and IMD in Table 1. The results have remained the same as before (p<0.001).

Comment 9: According to the Discussion section “The proportion of men undergoing their first test in the follow-up period increases steadily throughout the ten-year period” but I cannot find anywhere in the result section that calendar year has been explored.

Response: We agree that this sentence was slightly misleading and have removed it from the discussion.

Comment 10. Consider changing some of the references. Refer to the original source. For example, refer to the specific guidelines and to an article which investigates PSA-testing patterns around the world and not Harvey et al. (ref #1) which is an article about TRUS and Prensner et al. (ref #2) which is about the next generation of biomarkers.

Response: Thank you for pointing these two references out to us. We have now amended them to the following:

Harvey et al. 2012 => Office for National Statistics. Cancer Registration Statistics, England: 2015. <https://www.ons.gov.uk/peoplepopulationandcommunity/healthandsocialcare/conditionsanddiseases/bulletins/cancerregistrationstatisticsengland/2015> (accessed 7 July 2017)
 Presner et al. 2012 => Roobol M.J. The prostate-specific antigen test. Expert Opinion on Medical Diagnostics, 7(5), 423-426

References used in responses:

- *. Moss S, Melia J, Sutton J, et al. Prostate-specific antigen testing rates and referral patterns from general practice data in England. *Int J Clin Pract* 2016; 70(4):312-18.
- ** .Melia J, Moss S, Johns L, et al. (2004) Rates of prostate-specific antigen testing in general practice in England and Wales in asymptomatic and symptomatic patients: a cross-sectional study. *BJU International*; 94(1):51-6.
- ***. Roehrborn CG, McConnell JD, Lieber M, Kaplan S, Geller J, Malek GH, Castellanos R, Coffield S, Saltzman B, Resnick M, Cook TJ, Waldstreicher J, Grp PS (1999) Serum prostate-specific antigen concentration is a powerful predictor of acute urinary retention and need for surgery in men with clinical benign prostatic hyperplasia. *Urology* 53(3): 473-480.
- ****. National Institute for Health and Care Excellence [NICE] (2010) Lower urinary tract symptoms in men: management. In NICE Guidance. Available at: <https://www.nice.org.uk/guidance/cg97/chapter/1-Recommendations> (Accessed 14 March 2017).
- *****. Herrett E, Gallagher A M, Bhaskaran K et al. 2015. Data Resource Profile: Clinical Practice Research Datalink (CPRD). *International Journal of Epidemiology*. 44(3):827-836.
- *****. Hamdy F.C. Donovan J.L. Lane J.A et al. 2016. 10-year outcomes after monitoring, surgery, or radiotherapy for localized prostate cancer. *N Engl J Med* 2016; 375:1415-1424.

VERSION 2 – REVIEW

REVIEWER	Rebecka Arnsrud Godtman Department of Urology, Institute of Clinical Sciences, Sahlgrenska Academy at the University of Göteborg, Sweden
REVIEW RETURNED	24-Jul-2017

GENERAL COMMENTS	The authors have done a good job of answering almost all of the comments. However, one comment, regarding the missing PSA levels, has not been adequately addressed. “Those known to have had a PSA test but no level recorded were assumed to have undetectable levels (<0.1) and therefore equivalent to zero.” I agree that this assumption has a marginal effect on the results but the assumption does not make sense. The only men who would have undetectable PSA levels are those who have been treated for prostate cancer and since all men who had a prostate cancer diagnosis were excluded at start I find this assumption incorrect. Please change.
---

VERSION 2 – AUTHOR RESPONSE

Thank you for reviewing this manuscript. As requested, we have now removed the assumption that missing PSA levels are equivalent to zero. This has led to small alterations to the text in the methods and results, the footnote of Table 2, Table 3 and supplementary table S2.